# Audio-driven 3D Conversational Full-body Human Avatar Generation from a Single Image

## Abstract

Prior conversational 3D avatar systems require mapping audio to parametric poses and then pass through rendering pipeline. This forms a lossy bottleneck and introduces cumulative errors at the the pose–to–render interface, where quantization, retargeting, and per-frame tracking errors accumulate. As a result, they struggle to maintain tight audio–motion synchronization and to express micro-articulations crucial for conversational realism—bilabial closures, cheek inflation, nasolabial dynamics, eyelid blinks, and fine hand gestures—issues that are amplified under single-image personalization. We address these limitations with an end-to-end framework that constructs a full-body, photorealistic 3D conversational avatar from a single image and drives it directly from audio, bypassing intermediate pose prediction. The avatar is represented as a particle-based deformation field of 3D Gaussian primitives in a canonical space; an audio-conditioned dynamics module produces audio-synchronous per-particle trajectories for face, hands, and body, enabling localized, high-frequency control while preserving global coherence. A splat-based differentiable renderer maintains identity, texture, and multi-view realism, and we further enhance synchronization and natural expressivity by distilling priors from a large audio-driven video diffusion model using feature-level guidance and weak supervision from synthetic, audio-conditioned clips. End-to-end training lets photometric and temporal objectives jointly shape the audio-conditioned deformation and rendering. Across diverse speakers and conditions, our method improves lip–audio synchronization, fine-grained facial detail, and conversational gesture naturalness over pose-driven baselines, while preserving identity from a single photo and supporting photorealistic novel-view synthesis—advancing accessible, high-fidelity digital humans for telepresence, assistants, and mixed reality.

## 1 Introduction

Building highly realistic and animatable 3D human avatars has been a central ambition in computer vision and graphics for decades. Beyond static reconstruction, recent work increasingly targets controllable, identity-preserving 3D avatars driven by external signals—e.g., pose, audio, or driving video—with photorealistic novel-view synthesis. (Bagautdinov et al., 2021; Martinez et al., 2024; Ng et al., 2024; Zielonka et al., 2025; Agrawal et al., 2025). We refer to 3D animatable avatar as a personalized model that encodes a subject's canonical shape and appearance, deforms coherently under a driving signal, and photorealistic rendering. Despite rapid progress in neural rendering and learned deformation, two capabilities remain underexplored in combination: personalizing a full-body avatar from a single image, and expressing conversational talking motion directly from audio. Achieving both in a user-friendly pipeline is challenging because it requires recovering identity and deformation readiness from minimal input, and aligning subtle audio-conditioned dynamics across face, hands, and body at high temporal precision.

Template-based pipelines fit SMPL/SMPL-X (Loper et al., 2023; Pavlakos et al., 2019a), learn canonical geometry/texture, and drive them with pose-dependent LBS (Lewis et al., 2023), often coupled with NeRF or 3D Gaussians (Mildenhall et al., 2021; Kerbl et al., 2023), yielding photorealistic results across poses/views. However, they struggle with audio-synchronized conversational behavior—where millisecond lip closures, coarticulation, and fine hand gestures strongly affect naturalness since they require separate audio-to-motion module (Chhatre et al., 2024; Liu et al., 2024b; Mughal et al., 2025) that predicts parametric body/face/hand poses to drive a pose-conditioned renderer,

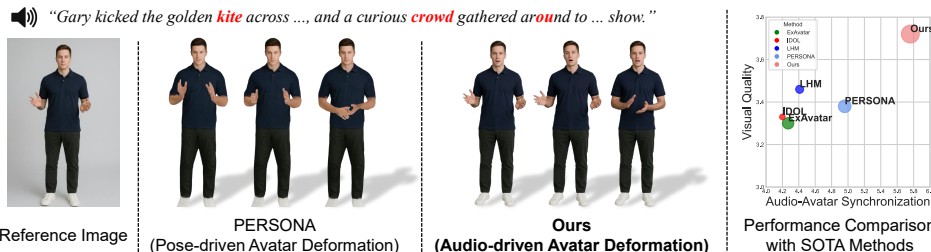

*"Gary kicked the golden **kite** across ..., and a curious **crowd** gathered ar**ou**nd to ... show."*

Reference Image | PERSONA (Pose-driven Avatar Deformation) | **Ours (Audio-driven Avatar Deformation)** | Performance Comparison with SOTA Methods

Figure 1: **Motivation**. When animating a 3D avatar with conversational motion from audio, state-of-the-art pose-driven deformation approach degrades visual quality (including facial expressions), yield less natural motion, and exhibit poor audio–motion synchronization. In contrast, our method directly controls the avatar from the audio signal, yielding substantial improvements in visual quality, motion naturalness, and synchronization. The table in the top-right reports a performance comparison under a single-image input setting across 3D-avatar baselines; circle markers denote motion naturalness, where larger circles indicate more realistic motion. For each method, we show the rendered frames aligned to the highlighted words in the driving audio.

where this introduces a lossy bottleneck, failing to capture tongue–lip contacts, cheek inflation, nasolabial detail, finger nuance and frame-by-frame deformation with weak temporal constraints, leading to sync errors 1. These issues intensify under single-image personalization, where recovering a deformation-ready canonical avatar and learning an expressive, audio-aligned controller from one photo is especially ill-posed.

We address these challenges with an end-to-end pipeline that builds, from a single user image, a full-body 3D conversational avatar whose motion is driven directly by audio, where we modulate audio features to learn a dense deformation and fine appearance field inside differentiable avatar deformer and neural renderer that preserves identity and photorealism. It enhances temporal alignment by sequence-level rendering losses, allowing gradients to flow through time and synchronize deformations with speech prosody, rather than relying on per-frame pose tracking. Training on paired audio–video sequences enables the model to realize micro-articulations and coordinated face–hand–body dynamics without a lossy audio-to-pose bottleneck.

At the core of our approach is a particle-based deformation field embedded in a differentiable 3D Gaussian renderer. From a single user photo, we reconstruct a canonical, identity-preserving avatar and instantiate Gaussian particles that are dense over expressive regions (lips, eyelids, fingers) and sparse elsewhere for efficiency. Audio features directly modulate per-particle trajectories—without an intermediate parametric pose—so that micro-articulations at the mouth, eyes, and hands can be controlled locally while the body motion remains globally coherent. Running this control at audio-synchronous rates expresses both rapid transients (e.g., plosive closures) and longer prosodic movements (e.g., head nods, beat gestures) with precise timing. Regularizers on locality and spectrum curb jitter yet preserve the high-frequency components essential for intelligible articulation.

To strengthen synchronization and realism under the single-image regime, we distill audio–motion priors from a pretrained audio-driven video diffusion model. Diffusion features provide a sequence-level alignment signal that nudges our particle dynamics toward plausible coarticulation and conversational gesturing; in addition, synthetic audio-conditioned clips serve as weak supervision to diversify motion while keeping it synchronized to the same audio. Training is end-to-end: rendering losses propagate through time into the audio-conditioned deformation field, allowing the renderer and dynamics to co-adapt for tight audio–visual alignment while preserving identity and photorealistic appearance across novel views.

**Contributions.** (1) We propose an end-to-end, single-image pipeline that maps audio directly to a dense differentiable deformation field inside a Gaussian renderer, eliminating the lossy audio-to-pose and pose-to-render handoffs where quantization/retargeting/per-frame tracking errors accumulate, thereby reducing drift and improving temporal alignment. (2) We introduce a particle-based representation that affords localized, high-frequency facial/hand control with globally coherent full-body motion, yielding precise conversational expressivity. (3) We develop a diffusion-distillation scheme

Table 1: Comparison of most related works for animatable 3D full-body avatar generation. Early works typically required multi-view or monocular videos, while recent methods enable avatar creation from a single image. However, most focus on general body motion rather than explicitly modeling co-speech gestures for talking avatars. Even approaches addressing talking avatars often rely on intermediate parametric pose conversion instead of directly driving avatars from audio, which prevents temporal deformation that enforces alignment with speech. Our method uniquely supports single-image input, full-body output, direct audio-driven control, explicit talking avatar generation, and temporally aligned deformation.

| Method | Input: Single-img. | Output: Full-body | Audio Driving | Talking Avatar | Temporal Deform. |
|---|---|---|---|---|---|
| ExAvatar (Moon et al., 2024) | ✗ | ✓ | ✗ | ✗ | ✗ |
| One-shot, One-talk (Xiang et al., 2024) | ✓ | ✓ | ✗ | ✓ | ✗ |
| IDOL (Zhuang et al., 2025) | ✓ | ✓ | ✗ | ✗ | ✗ |
| TaoAvatar (Chen et al., 2025a) | ✗ | ✓ | ✗ | ✓ | ✗ |
| AniGS (Qiu et al., 2025b) | ✓ | ✓ | ✗ | ✗ | ✗ |
| GUAVA (Zhang et al., 2025) | ✓ | ✗ | ✗ | ✓ | ✗ |
| LHM (Qiu et al., 2025a) | ✓ | ✓ | ✗ | ✗ | ✗ |
| PERSONA (Sim & Moon, 2025) | ✓ | ✓ | ✗ | ✗ | ✗ |
| **Ours** | ✓ | ✓ | ✓ | ✓ | ✓ |

that transfers audio–motion priors via feature alignment and synthetic audio-conditioned clips, enabling realistic, well-synchronized behavior with minimal personalization data.

# 2 RELATED WORK

## 2.1 ANIMATABLE 3D FULL-BODY HUMAN AVATARS

Early systems reconstructed actors from 3D capture or multi-view studios and animated the resulting meshes via hand-crafted pipelines—artist-designed rigging and skinning (e.g., LBS/DQS) or low-dimensional, PCA-based template models (Stoll et al., 2010; Alldieck et al., 2018; Joo et al., 2015; Pons-Moll et al., 2017; Habermann et al., 2019; Loper et al., 2023; Pavlakos et al., 2019b; Romero et al., 2022; Li et al., 2017). Pose-parameterized articulation enabled cross-subject transfer, but heavy expert intervention made these pipelines costly and time-consuming.

The advent of continuous implicit representations ushered in neural renderers such as NeRF (Mildenhall et al., 2021), powering photorealistic avatars (Peng et al., 2021b;a; Zheng et al., 2023; Shen et al., 2023; Su et al., 2021; Li et al., 2022; Wang et al., 2022) and free-view synthesis (Kwon et al., 2021; 2024; Weng et al., 2022; Guo et al., 2023; Liu et al., 2021). Yet NeRFs often train and infer slowly and need additional structure for reliable driving and retargeting. Acceleration via multi-resolution hash encodings and 3D Gaussian Splatting delivers real-time rendering with high-fidelity textures (Jiang et al., 2023; Kerbl et al., 2023), though many methods still rely on multi-view capture (Li et al., 2024; Pang et al., 2024) or monocular motion-capture signals (Moreau et al., 2024; Lei et al., 2024; Qian et al., 2024; Hu et al., 2024a; Moon et al., 2024; Guo et al., 2025; Hu et al., 2024b) rather than commodity monocular inputs. Complementary lines leverage video diffusion to obtain animatable avatars from a single image, achieving view-consistent appearance even with limited data (Sim & Moon, 2025; Xiang et al., 2024).

Motivated by these observations, we pursue high-quality conversational full-body avatars that reduce dependence on pose-template intermediates. Our approach couples implicit motion-based deformation with a particle-based deformation layer designed to retain fine facial dynamics and finger gestures, while remaining compatible with efficient neural rendering. This hybrid control aims to preserve expressiveness and temporal coherence under realistic driving signals, closing the gap between head-only audio-driven animation and fully articulated, photorealisticistic human avatars.

## 2.2 HUMAN VIDEO DIFFUSION MODELS

Video diffusion models (Wan et al., 2025; Blattmann et al., 2023) have become strong backbones for human video synthesis, enabling pose-guided animation from keypoints, dense or parametric poses (Zhang et al., 2024; Xu et al., 2024; Hu, 2024; Xia et al., 2024; Zhu et al., 2024; Tu et al., 2024). While these methods yield temporally consistent motion, they largely focus on coarse body animation and require audio-to-motion conversion. More recent large audio-driven diffusion models (Meng et al., 2025; Wang et al., 2025; Gan et al., 2025; Chen et al., 2025b; Tu et al., 2025) generate talking

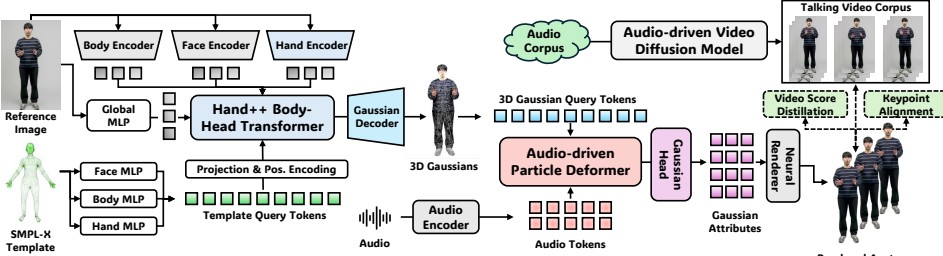

Figure 2: **Architecture overview of the proposed audio-driven 3D full-body avatar synthesis**. Given text or speech, we obtain audio features via TTS or an audio encoder and fuse them with template 3D-Gaussian query tokens to form driving tokens. A motion head and a Gaussian head—augmented by personalized LHM-Gesture++ priors and Face/Body/Hand MLPs—predict motion and appearance of 3D Gaussian particles with identity-adaptive skinning and linear blend skinning; a Gaussian decoder and neural renderer then produce the rendered avatar sequence. Training uses talking-video corpora with video score distillation and keypoint alignment, while projection and positional encoding bridge audio to geometry; an audio-driven particle deformer refines dynamics for natural lip–hand coordination. Inference reuses the same pathway from audio/text to motion & Gaussian tokens to generate photorealisticistic, synchronized talking videos.

videos directly from a single reference image and audio, producing realistic lip motion and gestures. However, they are typically limited to head/upper-body, rely on handcrafted or ground-truth guidance, operate at modest resolution, and struggle with fine-grained hand and facial details as well as identity preservation.

In contrast, our approach constructs a full-body audio-driven 3D avatar from a single image, overcoming the scope and fidelity limitations of prior work. By synthesizing diverse identity-specific talking videos from one image and varied audio, we enrich supervision for robust identity retention. Moreover, by distilling motion priors from large audio-driven diffusion models, our method achieves consistent coordination across body, hands, and face—capturing nuanced gestures and dynamic appearance beyond what existing approaches can deliver.

## 3 METHOD

**Overview.** Our goal is to build a personalized, whole-body conversational 3D avatar from a *single* image and to drive its face, hands, and body directly from audio at inference. Fig. 2 summarizes the pipeline. We first fine-tune a large human reconstruction model (LHM) (Qiu et al., 2025a) to the target subject (Sec. 3.1), augmenting it with a hand-enhancement branch; the model outputs a canonical avatar represented by 3D Gaussian particles. Projected query tokens are then processed by the Audio-driven Particle Deformer (Sec. 3.2) to produce audio-aligned deformation tokens, which Gaussian heads convert into deformed Gaussian attributes, rendered via neural splatting. To learn conversational dynamics from a single image, we distill a large audio-driven video diffusion teacher (Sec. 3.3) using *video score distillation* and *dense keypoint alignment*, alongside RGB and perceptual losses. At inference, a given audio sequence directly yields a rendered avatar video.

### 3.1 PERSONALIZING LHM FOR A CONVERSATIONAL AVATAR

**Baseline.** We adopt a large human reconstruction model (LHM) (Qiu et al., 2025a) as our baseline to regress a canonical whole-body avatar from a single input image and a coarse body prior. The avatar is represented by a set of 3D Gaussians $\mathcal{G} = \{(\boldsymbol{\mu}_i, \boldsymbol{\Sigma}_i, \alpha_i, \mathbf{c}_i)\}_{i=1}^{N}$, where $\boldsymbol{\mu}_i \in \mathbb{R}^3$ is the mean, $\boldsymbol{\Sigma}_i \in \mathbb{R}^{3 \times 3}$ the covariance, $\alpha_i \in [0, 1]$ the opacity, and $\mathbf{c}_i$ the view-conditioned color. A splatting-based rasterizer $\mathcal{R}$ renders photorealistic images in the canonical space, upon which our method builds.

**Enhancing Hand Representation.** Vanilla LHM focuses on general body motion and struggles with fine hand gestures that are crucial for conversational expressiveness. To specialize LHM for talking avatars, we personalize its multi–modal Body–Head Transformer by adding a dedicated hand–

attention branch (Hand++ Body–Head Transformer). Concretely, we extract hand–specific visual tokens using a ViT–based, pre–trained hand encoder $E_{\text{hand}}$ (Pavlakos et al., 2024) applied to hand RoIs. Let $\mathbf{Q}$ denote the query tokens from the LHM backbone and $(\mathbf{K}_{fb}, \mathbf{V}_{fb})$ the keys/values from the original face–body streams. We augment keys and values with hand tokens: $(\mathbf{K}_{\text{hand}}, \mathbf{V}_{\text{hand}}) = \text{Proj}(E_{\text{hand}}(\text{RoI}_{\text{hand}}))$; $\mathbf{K} = [\mathbf{K}_{fb} \| \mathbf{K}_{\text{hand}}]$, $\mathbf{V} = [\mathbf{V}_{fb} \| \mathbf{V}_{\text{hand}}]$, and compute cross-attention. This injects hand cues into the shared latent, enhancing geometry and fine-detailed hand appearance.

**Gaussian Decoder.** From the hand–enhanced Transformer features, a Gaussian decoder $D_{\mathcal{G}}$ outputs $\mathcal{G}$. Altogether with Hand++ Body–Head Transformer, we person–specifically fine–tune $D_{\mathcal{G}}$ to improve identity preservation while maintaining rendering stability.

## 3.2 AUDIO-DRIVEN PARTICLE DEFORMER

Template-driven LBS (Lewis et al., 2023) from SMPL-X (Pavlakos et al., 2019a) captures body motion well but under-expresses speech-synchronous micro-dynamics of the face and co-speech hand gestures. We therefore introduce an audio-driven particle deformer that converts acoustic/linguistic cues into time-varying deformations of implicit particles bound to the avatar's 3D Gaussian primitives. The module includes frame-synchronous audio embeddings from an audio encoder, text embeddings from a speech-to-text model, and avatar-conditioned 3D Gaussian query tokens. It outputs per-Gaussian residual updates produced by a *Gaussian Head*. These outputs are rendered by a splatting-based neural renderer.

**Audio/Text Encoders.** Given audio features $\mathbf{a}_{1:T}$, an audio encoder $E_{\text{aud}}$ (Baevski et al., 2020) produces $\mathbf{A}_t = E_{\text{aud}}(\mathbf{a}_t) \in \mathbb{R}^{d_a}$; optional transcripts are embedded by a text encoder $E_{\text{text}}$ (Radford et al., 2023) into $\mathbf{Z} = E_{\text{text}}(\text{text}) \in \mathbb{R}^{L_z \times d_z}$. We fuse modalities via a gated projector, using $\tilde{\mathbf{A}}_t = \text{PE}(t) \oplus \mathbf{A}_t$ and $\mathbf{D}_t = \gamma_t \text{Proj}_a(\tilde{\mathbf{A}}_t) + (1 - \gamma_t) \text{Pool}(\text{Proj}_z(\mathbf{Z}))$, where $\text{PE}(t)$ is positional encoding, $\oplus$ denotes concatenation, Pool aggregates over text/time, and $\gamma_t \in [0, 1]$ is predicted from $\tilde{\mathbf{A}}_t$. The fused token $\mathbf{D}_t \in \mathbb{R}^d$ serves as the driving signal at time $t$.

**Audio-Driven Particle Deformer.** We build our particle deformation module upon generative dynamics of 3D Gaussians (Xie et al., 2024). We define $M$ implicit particles $\mathcal{J} = \{j_m\}_{m=1}^M$, each producing an SE(3) transform $\mathbf{T}_{m,t} \in \text{SE}(3)$ per frame. Given *3D Gaussian query tokens* $\mathbf{Q} \in \mathbb{R}^{N_q \times d}$ from the personalized LHM (conditioned by face/body/hand encoders) and *template query tokens* $\mathbf{Q}_0$, we compute cross-attention to align speech cues with avatar structure:

$$\mathbf{K}_t = \text{Proj}_K([\mathbf{Q} \| \mathbf{Q}_0]), \quad \mathbf{V}_t = \text{Proj}_V([\mathbf{Q} \| \mathbf{Q}_0]), \tag{1}$$

$$\mathbf{H}_t^{\text{mot}} = \text{softmax}\left(\frac{\text{Proj}_Q(\mathbf{D}_t)\mathbf{K}_t^\top}{\sqrt{d}}\right)\mathbf{V}_t, \tag{2}$$

yielding motion-context features $\mathbf{H}_t^{\text{mot}} \in \mathbb{R}^{M \times d_h}$. A residual SE(3) parameterization is predicted as twist coordinates $\Delta\boldsymbol{\xi}_{m,t} \in \mathbb{R}^6$ and integrated via the exponential map:

$$\Delta\boldsymbol{\xi}_{m,t} = \text{MLP}_\xi(\mathbf{H}_t^{\text{mot}}[m]), \qquad \mathbf{T}_{m,t} = \exp(\widehat{\Delta\boldsymbol{\xi}_{m,t}}) \mathbf{T}_{m,t-1}, \tag{3}$$

with $\widehat{\cdot}$ the se(3) hat operator. The *Audio-driven Particle Deformer* and *Projection & Positional Encoding* blocks are shown in the architecture diagram.

**Gaussian Head (Appearance/Residual Geometry).** Speech induces fine nonrigid changes (lip rounding, teeth visibility, specular shifts). Complementary to LBS, the **Gaussian Head** predicts per-Gaussian residuals conditioned on both the driving token and Gaussian queries:

$$\mathbf{H}_t^{\text{gau}} = \text{CrossAttn}(\mathbf{D}_t, \mathbf{Q}), \tag{4}$$

$$\Delta\mathbf{p}_i(t) = \text{MLP}_\mu(\mathbf{H}_t^{\text{gau}}[i]), \quad \Delta\mathbf{s}_i(t) = \text{MLP}_\Sigma(\mathbf{H}_t^{\text{gau}}[i]), \tag{5}$$

$$\Delta\alpha_i(t) = \text{MLP}_\alpha(\mathbf{H}_t^{\text{gau}}[i]), \quad \Delta\mathbf{c}_i(t) = \text{MLP}_c(\mathbf{H}_t^{\text{gau}}[i]), \tag{6}$$

and applies them after motion:

$$\boldsymbol{\mu}_i^\star(t) = \boldsymbol{\mu}_i'(t) + \Delta\mathbf{p}_i(t), \tag{7}$$

$$\boldsymbol{\Sigma}_i^\star(t) = \boldsymbol{\Sigma}_i'(t) \oplus \Delta\mathbf{s}_i(t), \quad \alpha_i^\star(t) = \alpha_i + \Delta\alpha_i(t), \quad \mathbf{c}_i^\star(t) = \mathbf{c}_i + \Delta\mathbf{c}_i(t). \tag{8}$$

Here $\oplus$ denotes a stable covariance update. The *Gaussian Head* is shown alongside the *Motion Head* in the pipeline.

## 3.3 Distilling Audio-driven Large Human Video Difussion Model and Training

We leverage a audio-driven large human video diffusion model (Chen et al., 2025b) to enable a 3D avatar to express conversational motion from a single image. Specifically, we (i) augment person-specific talking video datasets (see supplementary for details), and (ii) transfer motion knowledge learned from large-scale data into the 3D avatar through a video score distillation objective. This also mitigates the identity preservation issues common when relying only on image-level losses. Below, we further describe video score distillation, dense keypoint alignment, and the total loss.

**Video Score Distillation.** Let $\mathbf{I}_{1:T}(\mathbf{\Phi})$ be frames rendered from our model parameters $\mathbf{\Phi}$ (particle deformer, motion/gaussian heads, skinning, renderer), conditioned on driving audio/text $c$. Denote the teacher score network $s_\psi(\cdot, \tau, c)$ at noise level $\tau$ with variance schedule $\alpha(\tau)$ and $\sigma(\tau)$. We apply a video variant of score-distillation sampling to inject the teacher's generative prior:

$$\nabla_{\mathbf{\Phi}} \mathcal{L}_{\text{vsd}} = \mathbb{E}_{t,\tau,\epsilon} \left[ w(\tau) \big( s_\psi(\mathbf{x}_{t,\tau}, \tau, c) - \epsilon \big) \frac{\partial \mathbf{x}_{t,\tau}}{\partial \mathbf{\Phi}} \right], \quad \mathbf{x}_{t,\tau} = \alpha(\tau) \, \mathbf{I}_t(\mathbf{\Phi}) + \sigma(\tau) \epsilon, \qquad (9)$$

which encourages $\mathbf{I}_{1:T}$ to lie on the teacher's audio-conditioned video manifold while inheriting its temporal coherence.

**Dense Keypoint Alignment.** To sharpen motion-phase alignment, we detect dense 2D face/hand keypoints from the teacher frames $\{\tilde{\mathbf{I}}_t\}$ and from our renderings $\{\mathbf{I}_t\}$. With $K_t^{\text{face}}$, $K_t^{\text{hand}}$ and their teacher counterparts $\tilde{K}_t^{\text{face}}$, $\tilde{K}_t^{\text{hand}}$, we minimize

$$\mathcal{L}_{\text{kpt}} = \sum_t \left[ \rho\big( \|K_t^{\text{face}} - \tilde{K}_t^{\text{face}}\|_2 \big) + \lambda_{\text{hand}} \rho\big( \|K_t^{\text{hand}} - \tilde{K}_t^{\text{hand}}\|_2 \big) \right], \qquad (10)$$

where $\rho$ is a robust penalty to handle detector noise and occlusions.

**Image-Level Supervision and Total Loss.** We constrain appearance with per-frame RGB and perceptual losses, $\mathcal{L}_{\text{img}} = \sum_t \|\mathbf{I}_t - \tilde{\mathbf{I}}_t\|_1 + \lambda_{\text{perc}} \sum_t \|\phi(\mathbf{I}_t) - \phi(\tilde{\mathbf{I}}_t)\|_2^2$, where $\phi$ is a fixed visual encoder. The full objective is $\mathcal{L} = \lambda_{\text{vsd}} \mathcal{L}_{\text{vsd}} + \lambda_{\text{kpt}} \mathcal{L}_{\text{kpt}} + \lambda_{\text{img}} \mathcal{L}_{\text{img}} + \lambda_{\text{reg}} \mathcal{L}_{\text{reg}}$, where $\lambda_{\text{vsd}}, \lambda_{\text{pkt}}, \lambda_{\text{img}}, \lambda_{\text{reg}}$ is scaling parameters, $\mathcal{L}_{\text{reg}}$ is an ARAP (as-rigid-as-possible) regularizer on deformations to preserve local rigidity and stabilize the avatar, jointly optimizing all modules for identity preservation, speech synchrony, and temporal smoothness.

## 4 Experiments and Analysis

### 4.1 Experimental Setup.

**Dataset.** Due to the limited availability of publicly accessible conversational human videos paired with audio, we aggregate data from multiple sources. In total, we collect and process $\sim$15,000 videos drawn from the Seamless-Interaction dataset (Agrawal et al., 2025), the Casual Conversational dataset (Porgali et al., 2023), additional online sources, and our own in-house captures. All videos depict a single human subject engaged in natural conversation, exhibiting both speaking activity and accompanying conversational motions, and each video is temporally aligned with its corresponding audio track. For evaluation, we create identity-aware splits: 10% of the videos for each identity are held out as a test set, and the remaining videos are used for training. Unless otherwise specified, the first frame of each video serves as the reference image of each models, for that identity.

**Metrics.** We evaluate our approach from multiple perspectives, using a variety of evaluation metrics. We evaluate the visual and aesthetic quality by evaluating IQA and ASE using Q-align (Wu et al., 2023). We adopt SyncC and SyncD, introduced by (Prajwal et al., 2020), to quantify the synchronization accuracy between lip motion and the corresponding audio. To evaluate the preservation of facial identity, we compute the cosine similarity (CSIM) between facial features extracted from the reference image and those from the generated frames. We further assess gesture fidelity using the average keypoint distance, reporting the Hand Keypoint Confidence (HKC) and the Hand Keypoint Variance (HKV), defined as the average confidence score and standard deviation of detected hand keypoints. For low-level reconstruction fidelity, we report PSNR and SSIM (Wang et al., 2004)

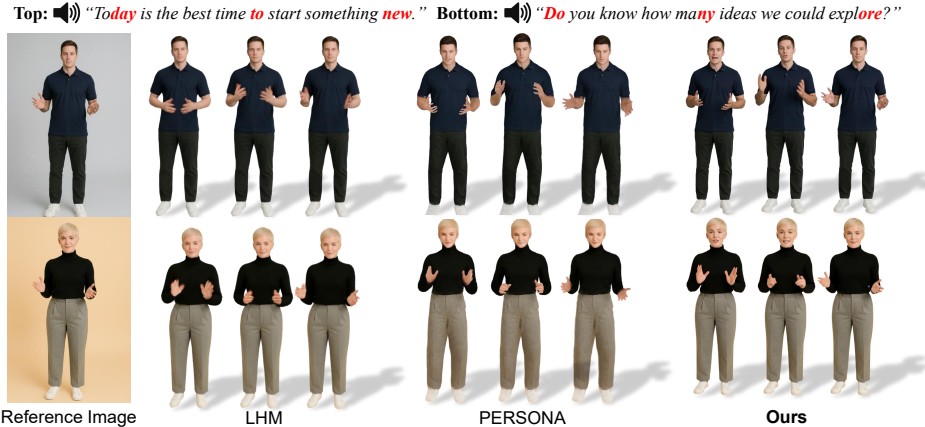

**Top:** 🔊 *"To**day** is the best time **to** start something **new**."*  **Bottom:** 🔊 *"**Do** you know how ma**ny** ideas we could expl**ore**?"*

Reference Image  LHM  PERSONA  **Ours**

Figure 3: **Qualitative comparison with state-of-the-art audio-driven large human video diffusion models.** For each method, we show the rendered frames aligned to the highlighted words in the driving audio. Our method outperforms state-of-the-art approaches in terms of visual quality, motion naturalness, and synchronization.

between the rendered images and the ground-truth video, given same audio driving signal. Lastly, we employ the Fréchet Inception Distance (FID) (Heusel et al., 2017) and the Fréchet Video Distance (FVD) (Unterthiner et al., 2019) to measure the generative diversity and overall coherence of rendered 3d avatars.

**Comparative Methods.**  For comparative analysis, we benchmarked our approach against the most relevant state-of-the-art methods for creating animatable 3D avatars from a single image, namely publicly available PERSONA (Sim & Moon, 2025) and LHM (Qiu et al., 2025a), through both quantitative and qualitative evaluations. However, unlike our approach, they cannot directly drive a 3D avatar from audio and therefore require a converter from audio to a sequence of SMPL-X pose parameters. To this end, we utilize a state-of-the-art whole-body motion converter (Bian et al., 2025) to generate motion, which is then used to control the 3D avatars of the baselines. In addition, since our framework incorporates a rendering pipeline capable of producing fully rendered videos, we further extended our comparisons to include several state-of-the-art audio-driven human video diffusion models, OmniAvatar (Gan et al., 2025) and HunyuanVideo-Avatar Chen et al. (2025b), to provide a broader evaluation.

### 4.2 RESULTS

**Quantitative Comparisons.**  Table 2 shows that our approach outperforms all baselines across the ten reported metrics on the test set. Against single-image 3D avatar methods, LHM (Qiu et al., 2025a) and PERSONA (Sim & Moon, 2025), our method achieves higher perceptual quality (IQA: +3.4%, ASE: +4.4%), better audio–lip synchronization (SyncC: +4.3%, SyncD: ↓20.3% vs. the best baseline), and stronger low-level fidelity (SSIM: +4.7%, PSNR: +4.3%). When compared with state-of-the-art audio-driven human video diffusion models, OmniAvatar (Gan et al., 2025) and HunyuanVideo-Avatar (Chen et al., 2025b), our method delivers markedly improved video-level realism and temporal coherence, reducing FID by 27.9% (12.4 vs. 17.2) and FVD by 25.0% (240 vs. 320) relative to the strongest baseline. We also observe consistent gains in hand–gesture fidelity (HKC: +2.5%), reflecting more reliable control of fine-grained motions. Overall, these results substantiate the effectiveness of our audio-driven 3D avatar pipeline, yielding robust improvements across perceptual, synchronization, reconstruction, and video-level metrics.

**Qualitative Comparisons.**  Fig. 3 qualitatively compares the baselines that synthesize animatable 3D avatars from a 3D image on the test sets. Because prior approaches cannot directly control a 3D avatar from audio, we evaluate rendering quality under the same motion for all methods to ensure a fair comparison. The results show that our approach produces sharper and more expressive

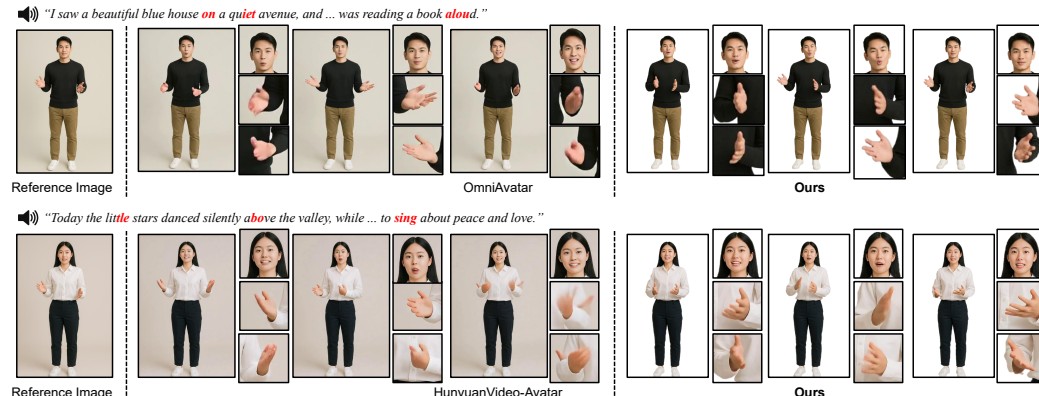

*"I saw a beautiful blue house on a quiet avenue, and ... was reading a book aloud."*

Reference Image    OmniAvatar    **Ours**

*"Today the little stars danced silently above the valley, while ... to sing about peace and love."*

Reference Image    HunyuanVideo-Avatar    **Ours**

Figure 4: **Qualitative comparison with human video generation models.** For each method, we show the rendered frames aligned to the highlighted words in the driving audio, along with cropped views of the *face* and *hands* for finer inspection. Relative to diffusion-based baselines, our approach exhibits fewer motion artifacts (e.g., lip–audio desynchronization, hand jitter/warping) and stronger identity preservation across views and phonetic contexts.

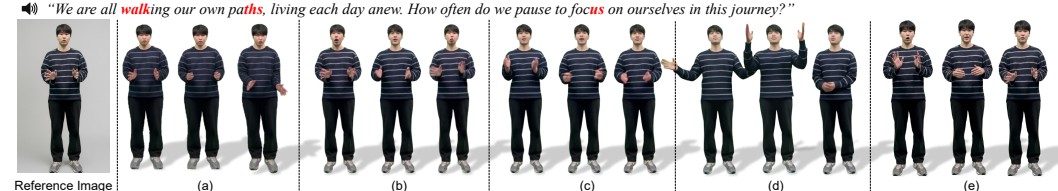

*"We are all walking our own paths, living each day anew. How often do we pause to focus on ourselves in this journey?"*

Reference Image    (a)    (b)    (c)    (d)    (e)

Figure 5: **Ablation of the proposed components.** We evaluate (a) w/o LHM personalization, (b) w/o $\mathcal{L}_{vsd}$, (c) w/o particle-based deformer, (d) w/o $\mathcal{L}_{kpt}$, and (e) full model. Removing any component harms visual fidelity, motion naturalness, and audio–motion sync, confirming each element's contribution to overall quality.

facial expressions, improved lip synchronization, and finer hand–gesture details. Across time, our renderings also exhibit smoother, more natural motion transitions.

Our rendering pipeline is built on a Gaussian rasterizer, enabling direct extraction of videos as sequences of images. We therefore also compare against state-of-the-art methods that generate human-animation videos from audio signals, as shown in Fig. 4. The visual comparisons indicate that our method achieves competitive image and motion quality even relative to recent generative video diffusion models. Notably, the second example highlights two consistent advantages of our approach: (i) motion-consistent preservation of fine hand details and (ii) stronger identity preservation throughout the sequence.

**Ablation Study.** We systematically ablate the proposed components and compare each variant to the full model across all metrics. Please refer to Table 3, and we qualitatively validate the proposed key components in Fig. 5.

*Personalization module.* Removing the fine-tuned Gaussian decoder (*w/o finetune. Gaussian decoder*) degrades perceptual and reconstruction quality (IQA and PSNR), while dropping the weight decoder (*w/o weight decoder*) increases distributional gaps (FID and FVD 275). Omitting the hand-enhancement pathway (*w/o hand enhancement*) notably reduces hand–gesture fidelity (HKC) despite otherwise moderate scores, confirming the role of high-frequency hand priors.

*Audio-driven particle deformer.* Excluding implicit motion tokens (*w/o implicit motion tokens*) harms audio–motion alignment (SyncC and SyncD) and raises video distances (FVD), indicating that compact motion cues are crucial for temporally coherent driving. Removing hand-gesture offsets (*w/o hand gesture offsets*) primarily impacts HKC, whereas removing facial-expression offsets (*w/o face expression offsets*) lowers perceptual quality and lip–face expressivity (IQA 4.00, SyncC 6.85).

Table 2: **Quantitative comparisons on the test set.** We compare our method with state-of-the-art methods that generate animatable 3D avatars from a single image and audio-driven human video diffusion models, across the evaluation metrics on the test set. Our approach consistently demonstrates significantly superior performance across all ten evaluation metrics.

| Methods | IQA↑ | ASE↑ | SyncC↑ | SyncD↓ | HKC↑ | CSIM↓ | SSIM↑ | PSNR↑ | FID↓ | FVD↓ |
|---|---|---|---|---|---|---|---|---|---|---|
| EchoMimicV2 (Meng et al., 2025) | 3.37 | 1.98 | 4.12 | 10.20 | 0.836 | 0.458 | 0.660 | 15.90 | 22.8 | 420 |
| OmniAvatar (Gan et al., 2025) | 3.99 | 2.64 | 6.40 | 7.60 | 0.858 | 0.525 | 0.705 | 17.20 | 18.6 | 350 |
| HunyuanVideo-Avatar (Chen et al., 2025b) | 4.08 | 2.71 | 6.90 | 7.12 | 0.875 | 0.539 | 0.709 | 17.55 | 17.2 | 320 |
| LHM (Qiu et al., 2025a) | 3.80 | 2.50 | 6.10 | 7.00 | 0.860 | 0.500 | 0.700 | 16.90 | 19.5 | 365 |
| PERSONA (Sim & Moon, 2025) | 3.88 | 2.58 | 6.30 | 6.80 | 0.868 | 0.510 | 0.708 | 17.20 | 18.9 | 345 |
| **Ours** | **4.22** | **2.83** | **7.20** | **5.42** | **0.897** | **0.551** | **0.742** | **18.30** | **12.4** | **240** |

Table 3: **Ablation study.** We demonstrate the effectiveness of our proposed components by systematically removing them and comparing against our full model across all evaluation metrics. The first block (rows 2–4) corresponds to the components introduced for personalizing large human reconstruction for conversational avatars. The second block (rows 5–7) includes the components introduced in the audio-driven particle deformer for the temporal deformation model. The third block (rows 8–9) consists of the objective functions incorporated to inject knowledge from audio-driven video diffusion models into our framework. The results highlight the importance of each proposed component, as all of them contribute to significant performance improvements across the evaluation metrics.

| Methods | IQA↑ | ASE↑ | SyncC↑ | SyncD↓ | HKC↑ | CSIM↓ | SSIM↑ | PSNR↑ | FID↓ | FVD↓ |
|---|---|---|---|---|---|---|---|---|---|---|
| w/o finetune. Gaussian decoder | 4.05 | 2.75 | 7.05 | 5.60 | 0.890 | 0.545 | 0.720 | 17.60 | 13.8 | 265 |
| w/o hand enhancement | 4.18 | 2.80 | 7.15 | 5.45 | 0.870 | 0.555 | 0.735 | 18.10 | 13.1 | 252 |
| w/o weight decoder | 4.10 | 2.72 | 6.95 | 5.70 | 0.885 | 0.540 | 0.728 | 17.80 | 14.5 | 275 |
| w/o implicit motion tokens | 4.08 | 2.70 | 6.60 | 6.10 | 0.882 | 0.538 | 0.726 | 17.70 | 15.2 | 300 |
| w/o hand gesture offsets | 4.16 | 2.78 | 7.10 | 5.50 | 0.860 | 0.552 | 0.734 | 18.00 | 13.7 | 258 |
| w/o face expression offsets | 4.00 | 2.74 | 6.85 | 5.80 | 0.888 | 0.520 | 0.730 | 17.90 | 14.2 | 272 |
| w/o video score distillation | 3.95 | 2.69 | 6.90 | 5.78 | 0.886 | 0.542 | 0.722 | 17.50 | 15.0 | 290 |
| w/o keypoint alignment | 4.02 | 2.73 | 6.70 | 6.20 | 0.872 | 0.544 | 0.725 | 17.60 | 15.6 | 310 |
| **Ours** | **4.22** | **2.83** | **7.20** | **5.42** | **0.897** | **0.551** | **0.742** | **18.30** | **12.4** | **240** |

*Objective functions.* Disabling video score distillation (*w/o video score distillation*) yields broad drops across perception and fidelity (IQA, ASE, FID, and FVD), and removing keypoint alignment (*w/o keypoint alignment*) produces the highest temporal distance (FVD) together with the worst sync error (SyncD), underscoring the value of geometry-aware supervision. Overall, the full model achieves the best results on all metrics, demonstrating that each component contributes meaningfully to perceptual quality, audio–lip synchronization, fine-grained gesture control, and temporal coherence.

## 5 CONCLUSION

In this paper, we proposed an end-to-end framework that constructs a personalized full-body 3D avatar from only a single image and drives its motion directly from raw audio. Unlike prior approaches that rely on intermediate parametric pose representations, our method eliminates the lossy audio-to-motion bottleneck and enables temporally precise, expressive conversational behavior. Leveraging a particle-based deformation model, the system captures fine-grained details in facial expressions and hand gestures while maintaining globally coherent body motion. Furthermore, by distilling motion priors from large-scale audio-driven video diffusion models, we enhance synchronization, motion diversity, and robustness under the single-image regime. Comprehensive experiments confirm that our framework delivers more photorealisticistic and synchronized talking avatars than existing baselines. We believe this formulation opens new possibilities for creating accessible, high-fidelity digital humans, with broad applications in telepresence, embodied AI, and immersive mixed reality environments.

**Ethics Statement.** This work makes use of publicly available datasets (Seamless-Interaction, Casual Conversational dataset) as well as a small amount of internally collected data. For all publicly available datasets, we adhere to their original license terms. For the internally collected data, explicit consent was obtained from the participants, and no personally identifying information beyond facial and vocal expressions was retained.

The proposed 3D talking avatar model has positive applications in telepresence, education, accessibility, and mixed reality systems. However, we acknowledge that the technology may be misused for harmful purposes, such as the creation of deceptive media. To mitigate such risks, we discuss limitations of the model and emphasize responsible use, including the potential integration of watermarking and detection mechanisms in deployment scenarios.

We also recognize possible concerns of fairness and bias, as datasets may not equally represent diverse demographics. We encourage future work to evaluate and expand the diversity of training data.

No sensitive personal information or medical data were used in this study. Institutional review board (IRB) approval was not required for the datasets employed, but ethical considerations regarding privacy, data protection, and informed consent were carefully followed.

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

# A    MORE RELATED WORKS AND DISCUSSION

**Co-Speech Gesture Video Generation.**    Similar to large-scale human video diffusion models, prior work has studied human video generation from audio, skeleton data, or 2D/3D poses, often through a two-stage pipeline: mapping audio to poses and then using a pre-trained GAN-based pose2video model (Ginosar et al., 2019; Qian et al., 2021). More recently, diffusion models (Hogue et al., 2024; Liu et al., 2024a; Qian et al., 2021; Liu et al., 2022; He et al., 2024) have been applied. A study (Huang et al., 2024) has also been introduced that generates a style-specific anchor avatar video from only a one-minute video clip. With notable work (Li et al., 2025) directly generating videos from audio, showing that bypassing the audio-to-pose step—long a performance bottleneck—can advance the task.

While this task shares the common goal of generating talking human videos from audio signals, it differs significantly from our approach: such methods typically produce only 2D videos rather than 3D avatars, and the generated content is limited to the upper body. Furthermore, they have been validated only on constrained domain-specific datasets, such as TED talks.

**Speech-driven Whole-body Motion Generation.**    This section focuses on methodologies that generate body, face, and hand parametric motions together. It is the task of automatically predicting natural, human-like body and hand gestures that align with spoken language. Unlike earlier works that focused on generating only facial expressions or body gestures in isolation, recent research has begun to explore the simultaneous generation of body, face, and hand gestures. These studies have introduced several methodological advances, including the use of VQ-VAE architectures (Yi et al., 2023), the adoption of large-scale datasets (Liu et al., 2024b), diffusion-based generative models (Chhatre et al., 2024; Chen et al., 2024; Mughal et al., 2025), and real-time generation (Liu et al., 2025) enabled by MAMBA or Flow Matching approaches. More recently, motion generation has been significantly improved through multi-task learning that incorporates diverse multimodal signals such as speech, text, and music, along with tasks including text-to-motion, audio-to-motion, and dance generation (Bian et al., 2025).

These methods map audio signals to co-speech gesture motions for 3D avatars, but their reliance on low-dimensional representations limits fine details such as wrinkles, facial nuances, and subtle hand gestures. Articulations and skinning on naked body meshes also cause deformation errors with clothed avatars. To address this, we propose an end-to-end pipeline that directly deforms 3D avatars from raw audio, reducing information loss and shape-induced errors while enabling photorealistic detail and expressive gestures. We further validated our approach through comparison with the state-of-the-art MotionCraft.

# B    ADDITIONAL RESULTS

We provide additional visual comparisons with methods that generate animatable 3D avatars from a single image, as well as with several audio-driven large human video diffusion models. Please refer to Fig. 6 and Fig. 7. Furthermore, we compare with One-shot and One-talk (Xiang et al., 2024), which are related works that generate 3D talking avatars from a single image. However, since their code and details regarding the train/test dataset split are not publicly available, it is difficult to conduct a quantitative comparison. Instead, we compare with the results released on their project pages to demonstrate the clear performance advantages of our approach. Please refer to Fig. 8.

# C    CONVERSATIONAL TALKING HUMAN DATASET CREATION

We describe in detail the pipeline for constructing a whole-body talking human video dataset paired with audio. An overview of the pipeline is illustrated in Fig. 9. We need to learn the whole-body motion of a 3D avatar from a single reference image. In this section, we leverage an audio-driven large human video diffusion model to achieve this goal. We propose a systematic pipeline for constructing a Conversational Talking Human Dataset, which integrates multimodal resources—text, audio, and video—to enable the generation of realistic, conversational human avatars. The process is designed to balance both diversity and consistency across different modalities, ensuring that the resulting dataset can serve as a strong foundation for speech-driven avatar generation and conversational AI research.

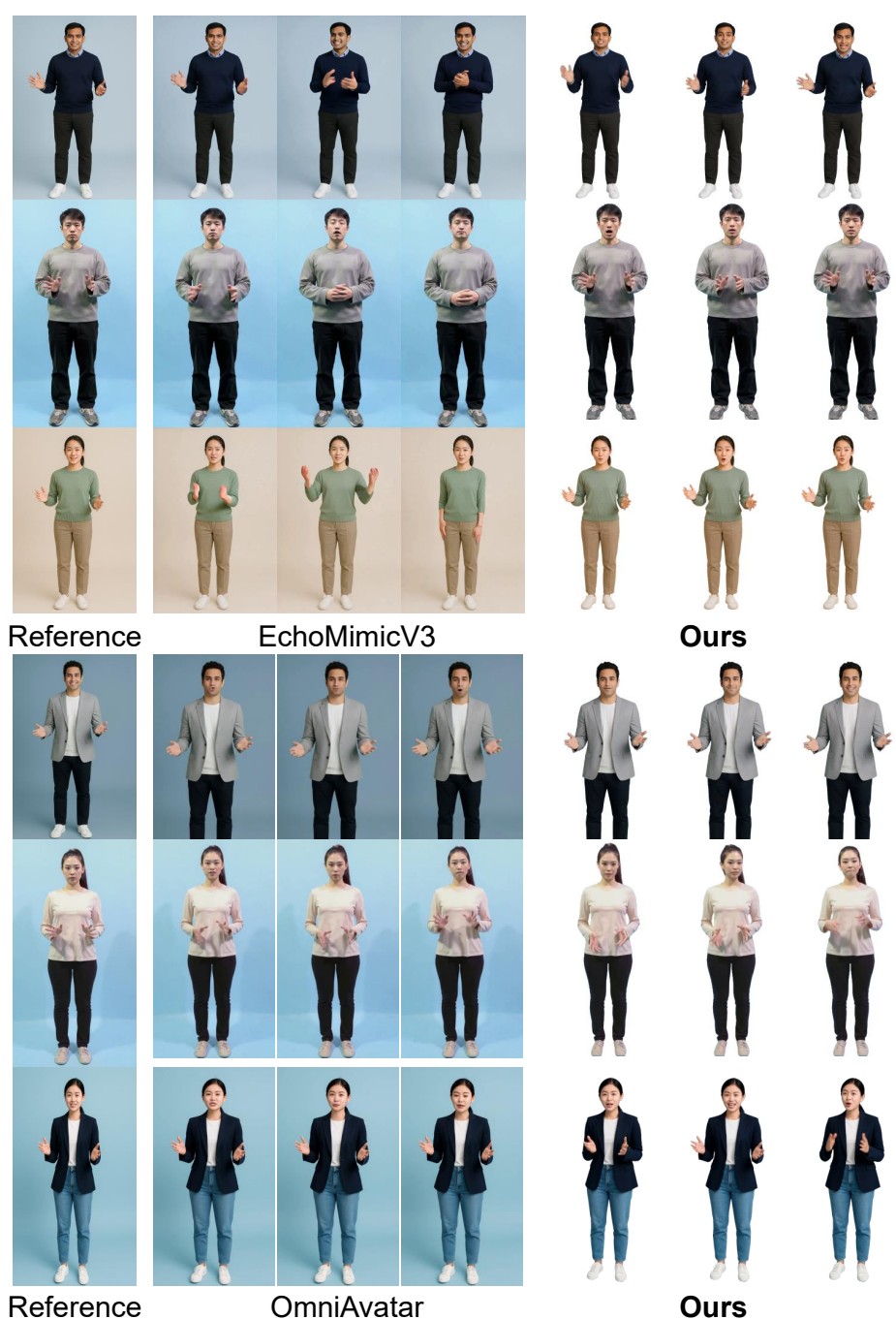

Figure 6: **Visual comparison with large human video diffusion models.** Our method shows improved identity preservation and reduced hand artifacts compared to human video diffusion models.

**Attribute Dictionary Construction.** The pipeline begins with the construction of an attribute dictionary that defines the diversity of the generated human figures. Attributes such as gender, age, body type, hairstyle, and clothing style are explicitly enumerated to create a wide spectrum of possible appearances. These attributes are embedded in carefully engineered text-to-image prompts that instruct the model to produce full-body renderings of realistic humans. Each generated image depicts a human making a conversational gesture, facing forward with both hands visible and

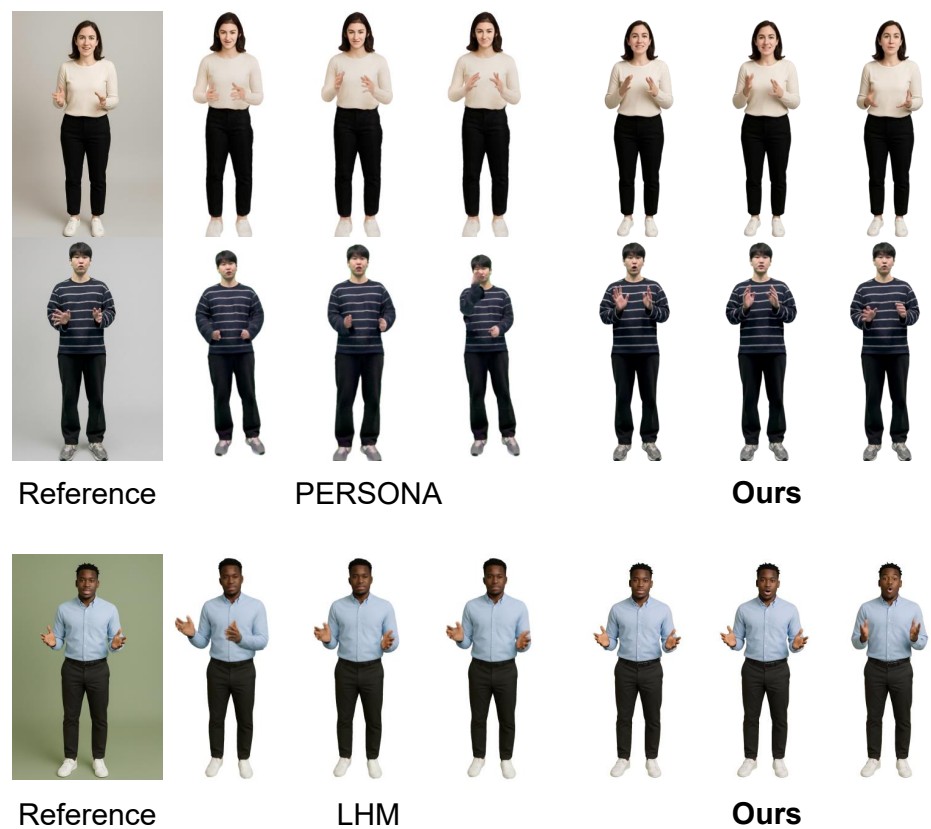

Figure 7: **Visual comparison with single-image animatable 3D avatars.** Our method shows superior visual quality and motion naturalness compared to them.

undistorted, while maintaining a solid background for visual consistency. This stage ensures not only diversity in representation but also structural integrity across the generated subjects.

**Text Corpus Design.** To simulate natural conversational dynamics, a dedicated text corpus is created. The corpus contains phrases that mirror authentic human interactions, including greetings, introductions, transitional statements, and engagement prompts. Rather than arbitrary text, these utterances are contextually grounded and resemble real dialogue or presentation scenarios. This design guarantees that the dataset captures the flow and tone of human-to-human communication. The collected text corpus is used as prompts at the LLM system (OpenAI, 2025).

**Speech Generation.** Each textual utterance is paired with a high-quality speech sample using text-to-speech (TTS) systems (ElevenLabs, 2025). Multiple variations in voice characteristics—including gender, timbre, and speaking style—are generated in order to reflect the natural diversity of spoken communication. This step enriches the dataset with acoustic variety and ensures that the resulting videos are not limited to a single vocal identity.

**Text-to-Image Human Generation.** Once the attributes and speech samples are defined, a diffusion-based text-to-image model (OpenAI, 2025) is employed to generate photorealistic human figures. By embedding the attribute specifications into prompts, the model produces visually consistent renderings that adhere to the conversational setting. Particular care is taken to enforce gestural realism, especially in the visibility and articulation of hands and fingers, as well as in the appropriateness of facial expressions aligned with conversational intent. The use of full-body images further enhances the realism and applicability of the dataset.

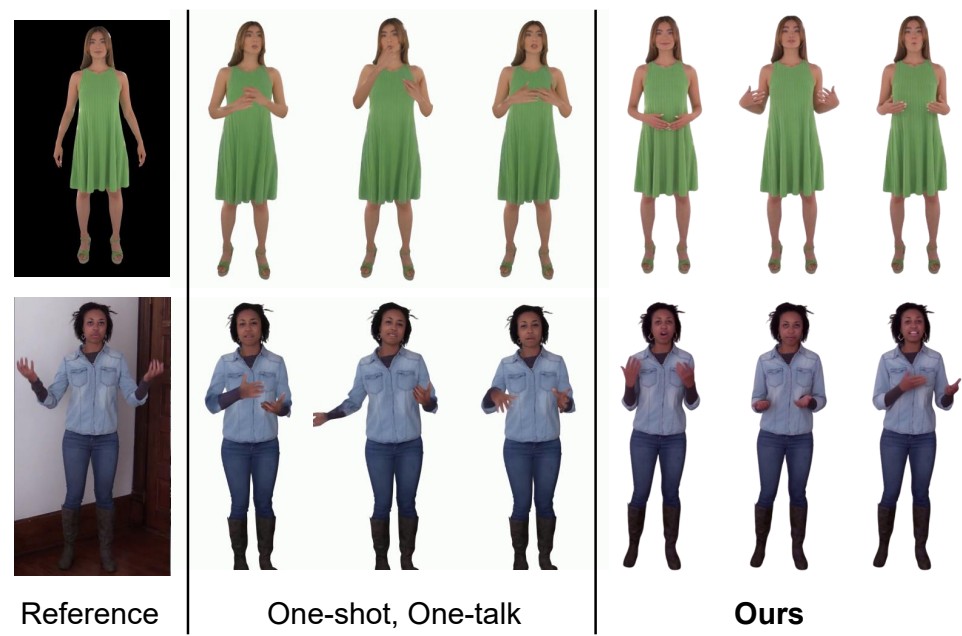

| Reference | One-shot, One-talk | **Ours** |

Figure 8: **Qualitative comparison with One-shot, One-talk (Xiang et al., 2024).** Since the code for these methods is not publicly available, quantitative comparison cannot be conducted. However, to demonstrate the superior performance of our proposed approach, we compare with the results released on the project page. Our method shows better hand appearance and gesture details. Moreover, it exhibits stronger ability to preserve facial identity across frames.

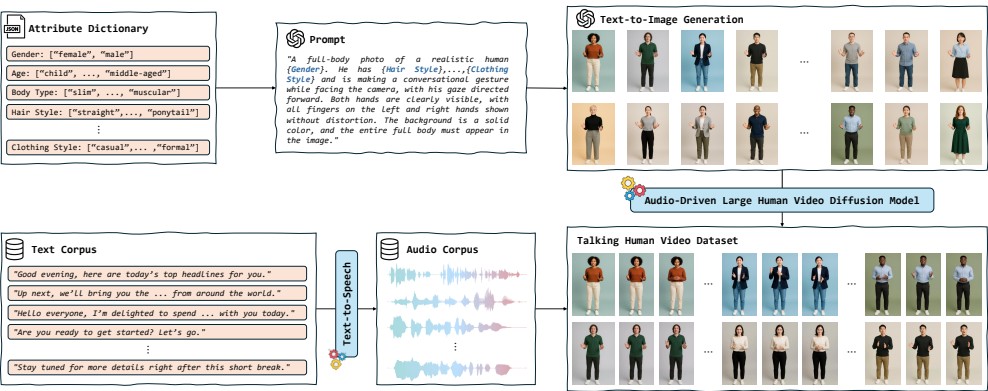

Figure 9: **Conversational Human Video Dataset creation pipeline overview.**

**Audio-Driven Talking Human Video Synthesis.** The crucial stage of the pipeline involves synchronizing static images with their corresponding audio through an audio-driven large human video diffusion model (Chen et al., 2025b). This model generates temporally coherent talking human videos by aligning lip movements, facial expressions, and subtle body gestures with the spoken content. The synthesis produces lifelike video segments in which the generated characters convincingly deliver the conversational utterances. The final dataset is assembled by systematically combining the generated human images, the conversational text corpus, the paired speech samples, and the synchronized talking human videos. This multimodal alignment provides a rich and diverse resource that can support applications, including realistic whole-body human avatar generation. The proposed dataset creation pipeline not only emphasizes diversity and naturalism but also ensures reproducibility and scalability for future studies in this field.

# D    LIMITATIONS

While our approach achieves compelling results, we acknowledge several limitations. **First**, **novel view synthesis** remains challenging. Because our system constructs avatars from a single input image and augments training data through audio-driven human video diffusion models, the generated samples are primarily near-frontal views, which limits performance under large viewpoint shifts. Nevertheless, our framework still produces high-fidelity avatars in typical front-facing scenarios, which are the most relevant for applications such as video conferencing, education, and digital assistants.

**Second**, our method does not yet support **interactive conversational avatar generation**. In natural conversations, gestures and expressions often adapt dynamically to the partner's speech and behavior, a factor not modeled in our current framework. Even so, by focusing solely on the speaker's audio, our method captures speech-synchronized motions with remarkable consistency, offering a reliable foundation for lifelike avatar animation. We see interactive modeling as an exciting avenue for future research, but our present approach already provides a strong step toward expressive and accessible human-avatar communication.

# E    BROADER IMPACTS

**Potential Negative Societal Impacts**   Our work advances high-fidelity, audio-driven 3D talking avatars but also carries risks. The technology could be misused to create deceptive or harmful media, such as deepfakes for misinformation, harassment, or identity fraud, raising ethical and legal concerns about trust in digital communication. Fairness and bias are also issues, as underrepresented groups in training data may experience degraded performance. Privacy risks emerge if avatars are generated without consent, and high computational demands may limit accessibility, reinforcing the digital divide.

**Broader Impacts**   At the same time, this technology offers significant benefits. Personalized 3D avatars can enhance telepresence, education, and remote collaboration, lowering communication barriers across diverse contexts. For individuals with disabilities, avatars may open new channels for expression and inclusion. The method also benefits entertainment, creative industries, and mixed reality applications. More broadly, it contributes to understanding the coupling of speech and gesture. To support responsible use, future work should incorporate safeguards such as watermarking, provenance tracking, and bias-aware evaluations.

