# OpenReview forum: "Audio-driven 3D Conversational Full-body Human Avatar Generation from a Single Image"
_ICLR.cc/2026/Conference — ICLR 2026 Conference Withdrawn Submission_

### Official Review · Reviewer_6aRf · 2025-10-30

**Soundness:** 3
**Presentation:** 4
**Contribution:** 3
**Rating:** 4
**Confidence:** 5

**Summary:**

This paper proposes an end-to-end framework for generating full-body, audio-driven 3D conversational avatars from a single image, bypassing traditional intermediate pose estimation. The method represents the avatar as a particle-based deformation field over 3D Gaussians, enabling direct audio-to-deformation control with fine-grained facial and hand articulation. It further leverages knowledge distillation from a large audio-driven video diffusion model to enhance motion naturalness and synchronization under the single-image setting. Experiments show improvements over pose-mediated 3D avatars and 2D diffusion baselines in metrics like lip-sync (SyncC/D), hand keypoint confidence (HKC), and video realism (FVD/FID).

While the architecture is technically sound and the results visually compelling, the paper overstates its 3D novelty (novel-view synthesis is not rigorously validated), lacks human evaluation, and does not isolate the contribution of synthetic data or the renderer from the core deformation design. Motion quality—especially gesture diversity and audio–gesture semantic alignment—is assessed only via indirect proxies. These gaps prevent a confident assessment of the method’s true advance over existing audio-driven avatar systems.

**Strengths:**

**Better cloth animation:**

The method avoids mesh-based skinning (e.g., SMPL-X LBS), which often causes cloth sliding or unnatural deformation. By using a particle-based deformation field over 3D Gaussians, it preserves fine surface details and avoids topology constraints. This leads to more natural cloth motion, especially around shoulders and torso. The renderer also retains texture fidelity from the single reference image.

**Good lip-sync compared to other full-body methods**

Quantitative results (SyncC↑, SyncD↓) and visual comparisons show tighter audio–lip alignment than pose-mediated baselines like LHM or PERSONA. This stems from bypassing the audio-to-pose bottleneck and using end-to-end temporal rendering losses. However, lip-sync is still evaluated only via automated metrics—not perceptual studies.

**End-to-end direct audio-to-deformation design**

The architecture eliminates intermediate pose estimation, reducing error accumulation and enabling micro-articulations (e.g., cheek puffing, eyelid blinks). The particle-based control allows localized high-frequency motion while maintaining global coherence. This is a meaningful architectural shift from template-driven pipelines.

**Weaknesses:**

**Lack detailed gesture quality analysis**
1. Lack of diversity analysis. Hand and body motions appear repetitive across utterances, lacking semantic variation (e.g., beat vs. iconic gestures). The method relies on distilled priors from a diffusion model, which may limit gesture vocabulary. No analysis is provided on gesture diversity, timing, or speech alignment.
2. Lack of human evaluation & user study. Claims about “naturalness,” “expressivity,” and “conversational realism” rely solely on proxy metrics (FVD, HKC) and visual inspection. Without MOS or A/B preference studies, it’s unclear if users perceive the motion as more realistic. This weakens the paper’s core user-facing claims.
3. Lack of analysis on audio–gesture relation. The paper does not test whether gestures are meaningfully tied to speech content (e.g., stress, semantics). A critical experiment—driving the avatar with mismatched audio—would reveal if gestures are audio-contingent or generic. Without this, gesture “naturalness” remains unverified.

**Novel-view synthesis: only fronts are shown**

All qualitative results are near-frontal; the paper admits poor performance under large viewpoint shifts (Appendix D). If the method cannot render consistent side/back views, its 3D advantage over 2D diffusion models is questionable for applications requiring true 3D interaction. No multi-view metrics (e.g., view SSIM, depth consistency) are reported. Although the author mentioned this in the limitation, but I'm still wondering if multi-view consistency can be achieved much easier than pure 2D baselines.

**Questions:**

**Can you handle complex cloths like long dresses?**

The current demos show only short sleeves or fitted clothing. Long, flowing garments involve complex dynamics (e.g., self-contact, wind interaction) that Gaussian primitives may not capture. It’s unclear if the particle density or deformation model scales to such cases.

**How’s the throughput?**

The paper omits FPS, latency, or real-time capability—critical for telepresence or AR/VR. While 3DGS enables fast rendering, the per-frame audio-driven deformation and autoregressive SE(3) updates may bottleneck inference. Without timing data, practical deployment remains uncertain.

**Other questions**
1. Identity preservation under motion: Does facial identity degrade during extreme expressions or gestures?
2. Audio input robustness: What happens with noisy audio, silence, or non-speech sounds?

---

### Official Review · Reviewer_ms4y · 2025-10-31

**Soundness:** 3
**Presentation:** 3
**Contribution:** 2
**Rating:** 6
**Confidence:** 4

**Summary:**

The paper introduces an end-to-end framework that constructs a full-body, realistic 3D avatar from a single image and drives it directly with audio. To address the loss of nuanced details (such as finger movements) common in template-based driving approaches, this work introduces 1) an end-to-end pipeline that maps audio directly to a deformation field inside a Gaussian renderer, circumventing error accumulation from intermediate pose estimation, 2) a particle-based representation for precise local control and globally coherent full-body motion, and 3) diffusion-distillation scheme that leverages priors for high-quality, synchronized animation with limited personalization data.

**Strengths:**

1.Unlike prior methods that rely on intermediate pose templates, this work introduces an end-to-end framework that directly drives a full-body 3D avatar from raw audio, overcoming the limited expressiveness of template-based approaches.
2.The adoption of a particle-based representation, which utilizes an implicit deformation field to model the motion of Gaussians, enhances the generation of high-frequency details in intricate areas such as the hands and face.
3.The diffusion-distillation scheme enables data-efficient learning of realistic, synchronized avatar behavior through feature alignment and synthetic clips.

**Weaknesses:**

1.The presented results show compelling performance on subjects with relatively simple textures and fitting clothing. It would be valuable to further explore the method's performance under more challenging conditions, such as complex patterns or loose garments, to understand its generalizability.
2.For a fair comparison with the LHM baseline, it would be helpful to clarify whether it was fine-tuned on the conversational dataset. A comparison against a fine-tuned LHM would more directly highlight the benefits of the proposed architecture.
3.The initial personalization stage involves fine-tuning the Hand++ Body-Head Transformer. It would be valuable to report the associated computational cost and time, as these are important factors for assessing the practical usability of the method.

**Questions:**

While the limitations note challenges in novel view synthesis, I would be interested to see more concrete evidence of its current performance, such as qualitative examples or a quantitative metric (e.g., PSNR from a held-out view).
I am also curious about whether fine-tuning the Gaussian decoder, compared to the original LHM, introduces a trade-off between enhancing front-view expressiveness and preserving the capability for accurate novel view synthesis.

---

### Official Review · Reviewer_TF4u · 2025-10-31

**Soundness:** 3
**Presentation:** 2
**Contribution:** 3
**Rating:** 4
**Confidence:** 4

**Summary:**

The paper proposes a method for audio-driven avatar generation from a single RGB  image.

To this end the authors leverage the pretrained LHM model, which can already predict 3D Gaussian attributes from a single RGB image, but is animated using SMPL pose conditions. The authors argue that using SMPL poses are missing important details to achieve realistically behaving conversation avatars, and that going from audio to SMPL-poses to avatar can accumulate errors.
Therefore, the authors propose to directly learn deformations of Gaussians from the audio condition.

In order to obtain more stable training, the authors extend the typical set of losses (RGB, perceptual, regularization), using distillation losses from a recent pre-trained, audio-conditioned video diffusion model. To be specific video score distillation is used to guide predictions in the direction of the video diffusion model. Moreover, a keypoint loss between keypoint detections of the predicted images and video-diffusion generations put important constraints on human movement.

To validate their approach the authors evaluate against 3D baselines, which are conditioned on SMPL poses and require an audio-to-SMPL estimator. Secondly, the authors compare against audio-conditioned video-diffusion models. The proposed method outperforms all baselines, and additionally several important design decisions are ablated to be beneficial.

**Strengths:**

Overall the paper seems to be focused on obtaining high-quality visual results. To this end a large pipeline leveraging two large-scale human-specific models are successfully used, and the resulting excellent visual quality of the proposed method is one of its main strengths.

Directly conditioning on audio instead of SMPL-pose is an important aspect, which is novel for whole body avatars. In the face domain, such a concept has been proposed previously, e.g. in VASA-1. Nevertheless, it seems to be crucial addition to the whole body domain, although it has not been perfectly ablated (see below).

With recent improvements on video-diffusion models, the distillation strategy is a useful idea that can be also be applied to related domains. As such, it could become a significant part of future work, due to its simplicity and effectiveness.

Most parts of the paper are well-written and easy to follow, except for the method section which is missing some details (see below)

**Weaknesses:**

Below I list some major and minor weaknesses in no particular order:
- (A) Neither the paper nor the video seems to show renderings from novel views. This raises the question whether the model still produces correct 3D geometry, or whether the renderings only look good from the same viewpoint as the input image. If that would be the case, one arrives at the question what the the benefit of a enforcing a 3D representation is. Learning to generate image from the same perspective might be easier when directly operating in 2D space,  e.g. how would the method compare to a fine-tuned version of HunYuan-VideoAvatar?

- (B) The evaluation seems to be heavily favoring the proposed method, since the training and evaluation is performed on the **same dataset** AND the **same identities**, i.e. only 10% of the frames are excluded. This heavily benefits the proposed model, since it can simply overfit the appearance of the identities, which constitutes large parts of the visual and perceptual metrics. But at least the hand gesture and synchronization metrics are less effected by this. Similarly, this also concerns the presented ablations, since it might favor methods with a larger representation capacity that can overfit better. Either I am somehow mistaken on this, or the evaluation on held-out IDs and ideally new datasets or in-the-wild examples (at least some qualitative comparisons) would be required to move the paper above the acceptance threshold. I am looking forward to hear the authors explanation for the chosen evaluation, and I am willing to upgrade my score based on it.


- (C) The method is not perfectly clearly described, especially when it comes to prediction of movement, which is on of the core technical novelties. E.g. the method section is not self-contained when it comes to the movement of gaussians, e.g. it is not explained what the predicted $T_{m,t}$ are used for, neither is $\mu^{\prime}$ defined, altough it is used in eq. (7). This should be improved for the final version of the paper.

- (D) On a semantic level, one of the papers core novelties is direct conditioning on audio compared to a combination of audio-to-SMPL and SMPL-conditioning. A proper ablation experiment for this would help the readers gauge the signifcance of this change. E.g. by using SMPL-based animation and predicting SMPL codes from audio.

**Questions:**

- What does neural renderer mean, is it not regular 3DGS rasterization anymore?
- Is the 3D consistency broken when training on this sort of data? This would be a core advantage compared to video-diffusion models, which are likely to scale better for audio conditioned 2D avatars.
- How exactly does the deformation work? it seems that both the "Particle Deformer" and the "Gaussian Head" can predict some sort of transformation/offset to move around the Gaussians.

---

### Official Review · Reviewer_6N8z · 2025-11-01

**Soundness:** 2
**Presentation:** 2
**Contribution:** 1
**Rating:** 4
**Confidence:** 3

**Summary:**

This paper proposes an end-to-end framework for generating a full-body, photorealistic 3D conversational avatar from a single image, driven directly by audio. The method bypasses traditional audio-to-pose pipelines by introducing an audio-driven particle-based deformation field operating over 3D Gaussian primitives. It integrates a differentiable neural renderer and leverages video score distillation from large-scale audio-driven diffusion models to enhance synchronization and realism.

**Strengths:**

End-to-end architecture: The proposed audio-driven particle deformer eliminates the lossy audio-to-pose bottleneck, a key limitation in previous avatar systems.

Single-image personalization: Good results given only one input photo, demonstrating robust identity preservation and realistic animation.

Integration with diffusion priors: The use of video score distillation from a large audio-driven diffusion teacher improves the quality of results from the baseline they start with.

**Weaknesses:**

- The paper claims it needs a single image to get the final result, but in the experiment section, it says they take 80% of the videos as training data.

- Scalability and efficiency: Although 3D Gaussian splatting improves speed, training and inference costs for full-body rendering are not clearly analyzed.

- Data dependence: The framework relies on distillation from large diffusion models trained on massive datasets, yet the proposed method's lip synchronization results are better than the diffusion model itself. It's not clear how this is possible

- Also, the paper claims in L185 that the video diffusion models struggle with preserving the identity, and yet use the same video model to distill knowledge to the proposed model, and claim better identity preservation. It's a bit of a contradictory loss.

**Questions:**

- How robust is the model to out-of-domain audio, such as emotional or non-conversational speech?

- Can the approach generalize to non-frontal or multi-person scenes?

- How does the temporal stability behave in long sequences (beyond 30 seconds)?

---

### Note · Authors · 2025-11-14

**Comment:**

We would like to formally withdraw our paper from the ICLR 2026 submission process. After careful and thorough consideration, we have decided to revise and further improve our work in light of new findings and the constructive feedback we have received. We believe that these revisions will allow us to make a stronger and more meaningful contribution in the future.

We sincerely appreciate the time, effort, and thoughtful evaluations provided by the reviewers and Area Chairs.

**Withdrawal Confirmation:**

I have read and agree with the venue's withdrawal policy on behalf of myself and my co-authors.